# The Analysis and Verification of IMT-2000 Base Station Interference Characteristics in the FAST Radio Quiet Zone

Jian Wang [1,2,3], Yibo Zhao [1], Cheng Yang [1,2,*], Yafei Shi [1,2,*], Yulong Hao [1], Haiyan Zhang [4,5,6], Jianmin Sun [7] and Dingling Luo [8]

1 School of Microelectronics, Tianjin University, Tianjin 300072, China; wangjian16@tju.edu.cn (J.W.); haoyulong98@tju.edu.cn (Y.H.)
2 Qingdao Institute for Ocean Technology, Tianjin University, Qingdao 266200, China
3 Shandong Engineering Technology Research Center of Ocean Information Awareness and Transmission, Qingdao 266200, China
4 National Astronomical Observatories of China, Chinese Academy of Sciences, Beijing 100101, China
5 National Key Laboratory of Radio Astronomy and Technology, Beijing 100101, China
6 Hebei Key Laboratory of Radio Astronomy Technology, Shijiazhuang 050081, China
7 Radio Monitoring Station of Guizhou, Guiyang 550001, China
8 Guizhou Province Network and Information Security Evaluation and Certification Center, Guiyang 550001, China; noah_leo@163.com
* Correspondence: ych2041@tju.edu.cn (C.Y.); shiyafei@tju.edu.cn (Y.S.)

**Abstract:** In this study, we aim to analyze the electromagnetic interference (EMI) regarding the Five-hundred-meter Aperture Spherical radio Telescope (FAST) caused by base stations in the International Mobile Telecommunications-2000 (IMT-2000) frequency band. By analyzing the frequency bands used by the transmitting and receiving devices and the surrounding environmental parameters and utilizing an approach to predicting radio wave propagation loss that is based on deterministic methods, we conclude by comparing the predicted received power at the FAST with its interference protection threshold. Our analysis demonstrates that, currently, only 55.31% of IMT-2000 base stations in the FAST radio quiet zone (RQZ) meet the protection threshold. Additionally, this article verifies the applicability and accuracy of the radio wave propagation model used in the research based on field strength measurements. Overall, this study provides valuable insights for improving the electromagnetic environment surrounding FAST and reducing the EMI caused by mobile communication base stations. It also provides corresponding analysis methods and useful suggestions for analyzing electromagnetic radiation interference in other radio telescopes.

**Keywords:** radio telescopes; electromagnetic interference; IMT-2000; FAST

## 1. Introduction

The Five-hundred-meter Aperture Spherical radio Telescope (FAST) was approved in 2007 and completed acceptance testing in 2020, officially entering its operational phase. As a major national advanced scientific and technological project in China, it holds the record for being the largest spherical radio telescope with a single-dish configuration and the most exceptional sensitivity worldwide [1–3]. FAST, known as the "Chinese Eye of the Sky", has a comprehensive performance about 10 times higher than that of the Arecibo telescope in the United States, which was destroyed in December 2020 [4]. Since its completion, FAST has achieved many significant scientific research results [5,6]. As a passive observational device for radio astronomy, radio telescopes rely on high-sensitivity terminal equipment to receive radio signals from celestial bodies for scientific research. Therefore, the radio astronomy service is highly vulnerable to electromagnetic interference (EMI) from other active services, which may affect its inherent scientific observations [7]. In addition, to strictly control the electromagnetic environment around FAST, the local government has established a radio quiet zone (RQZ) centered on the FAST site, with a radius of 30 km.

Among them, 0–5 km is the core area, 5–10 km is the middle area, and 10–30 km is the remote area. However, numerous mobile communication base stations have yet to meet the daily communication needs of local residents. These base stations may interfere with FAST during operation and affect its performance. Therefore, it is crucial to monitor the electromagnetic environment surrounding FAST, and exploring the signal radiation pattern of mobile communication base stations is essential to ensure the inherent observation of FAST.

Due to the ultra-high sensitivity of FAST, it is susceptible to electromagnetic signals. To mitigate interference from other businesses, it is necessary to implement electromagnetic compatibility and interference avoidance measures before construction and operation [8,9]. For example, Wang et al. have established a FAST satellite interference mitigation system for satellite interference sources. This system uses specific antennas to detect satellite interference in the 1–5 GHz frequency band and establishes a satellite EMI database to suppress satellite interference. Through practical observation, the feasibility of this system has been verified, and a practical solution has been provided to mitigate the impact of satellite interference on FAST [10]. Wang et al. analyzed the fading characteristics in the UHF band of the karst landforms around the FAST site based on the experimental datasets of six frequency channels around FAST using the Kolmogorov–Smirnov statistical method [11]. Additionally, based on radio propagation methods and cognitive theory, they also analyzed the radiation characteristics of mobile communication base stations in the 870–878.6 MHz frequency band within the FAST RQZ. Four strategies for interference avoidance and frequency coordination were proposed for base stations exceeding the FAST interference protection threshold, and the consistency of these strategies was verified through experimentation. In addition, a set of evaluation criteria for frequency coordination strategies was established to analyze the satisfaction of the proposed strategy under different criteria. This research provides theoretical and experimental support for improving the radiation interference of mobile communication base stations within the FAST RQZ [12].

In order to analyze the potential EMI caused by International Mobile Telecommunications-2000 (IMT-2000) mobile communication base stations on the operation of FAST, we use the method listed in Recommendation ITU-R. Based on the parameters of equipment and frequency information at the transceiver end, combined with the protection requirements of radio astronomy services, we analyze whether the IMT-2000 base station in the FAST RQZ will generate EMI to FAST.

## 2. Analysis Object

### 2.1. FAST

The location of FAST is situated in Kedu Town, Dawodang, within Pingtang County, located in the Guizhou Province of China (106.85° E, 25.64° N). With its natural geographic advantages and the unique structure designed by Chinese scientists, it is currently the world's largest and most sensitive single-dish radio telescope and a major scientific and technological infrastructure project of China's Eleventh Five-Year Plan. The scientific objectives of FAST mainly include: observing neutral hydrogen in the universe, searching for new pulsars, measuring very long baseline interference (VLBI), and extending deep space communication capabilities [13]. The FAST receiving and terminal systems currently include seven sets of receivers involving 70 MHz–3 GHz [14]. As shown in Figure 1, the IMT-2000 base stations around FAST include two operators, such as China Mobile Communications Corporation (CMCC) and China Unicom Communications Corporation (CUCC), mainly distributed within a range of 5–30 km from FAST, as radio equipment is prohibited in the core area of RQZ.

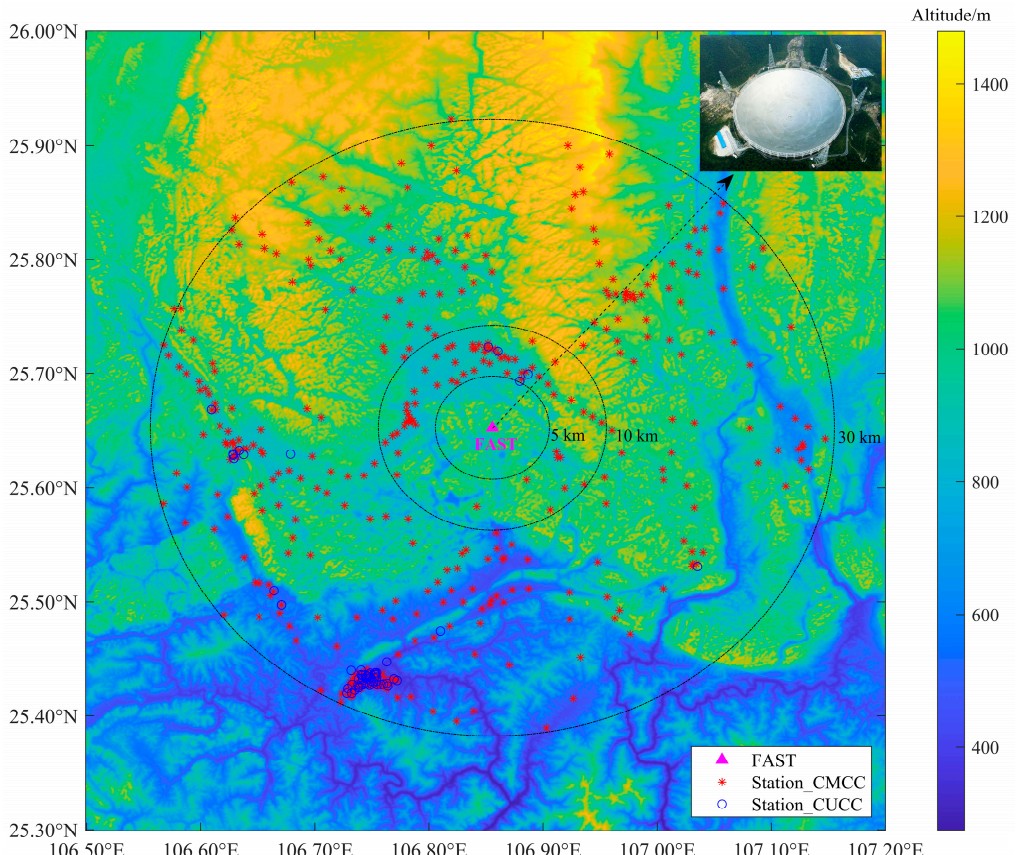

**Figure 1.** The distribution of IMT-2000 base stations around FAST.

According to the interference protection requirements for FAST and Recommendation ITU-R RA.769 [15], IMT-2000 base stations mainly involve two types of receivers in FAST, namely, B05 and B07. The operating frequency range of the B05 receiver is 1100–1900 MHz, and the system noise temperature is 20 K, while the operating frequency range of the B07 receiver is 2000–3000 MHz, and the system noise temperature is also 20 K. The relevant interference threshold values are shown in Figure 2. Furthermore, in the FAST-IMT interference analysis, we are particularly concerned with the maximum input power that the FAST can tolerate, and the B05 and B07 receiver protection power thresholds are both −199 dBW.

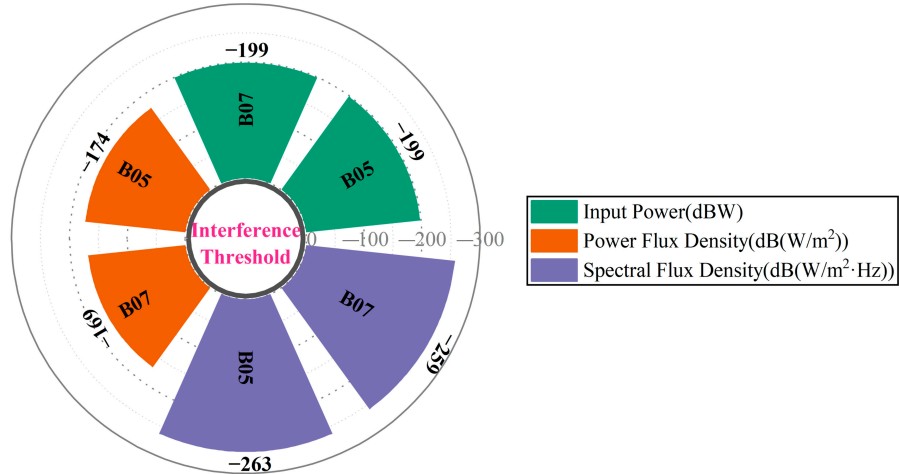

**Figure 2.** The interference protection requirements for FAST.

In analyzing interference on the FAST-IMT link, the FAST antenna serves as the receiving end, and its antenna gain in the direction of the received signal is the primary consideration. The typical antenna radiation patterns of the FAST antenna include the main vertical plane (zenith angle is 0°), the central horizontal plane (elevation angle is 0°), and the three-dimensional radiation pattern, as shown in Figure 3. The pattern of the antenna reveals a well-shaped main lobe with a narrow beamwidth, with a peak gain of 74.21 dB, accompanied by a number of minor lobes and back lobes of lower gain, with the highest side lobe located at about 120° with respect to the main lobe, as shown in Figure 3a. Figure 3b shows that the horizontal pattern of FAST is directionless over 360° when the elevation angle is 0°, exhibiting a uniform distribution of equivalent radiated power with a gain of 74.21 dB.

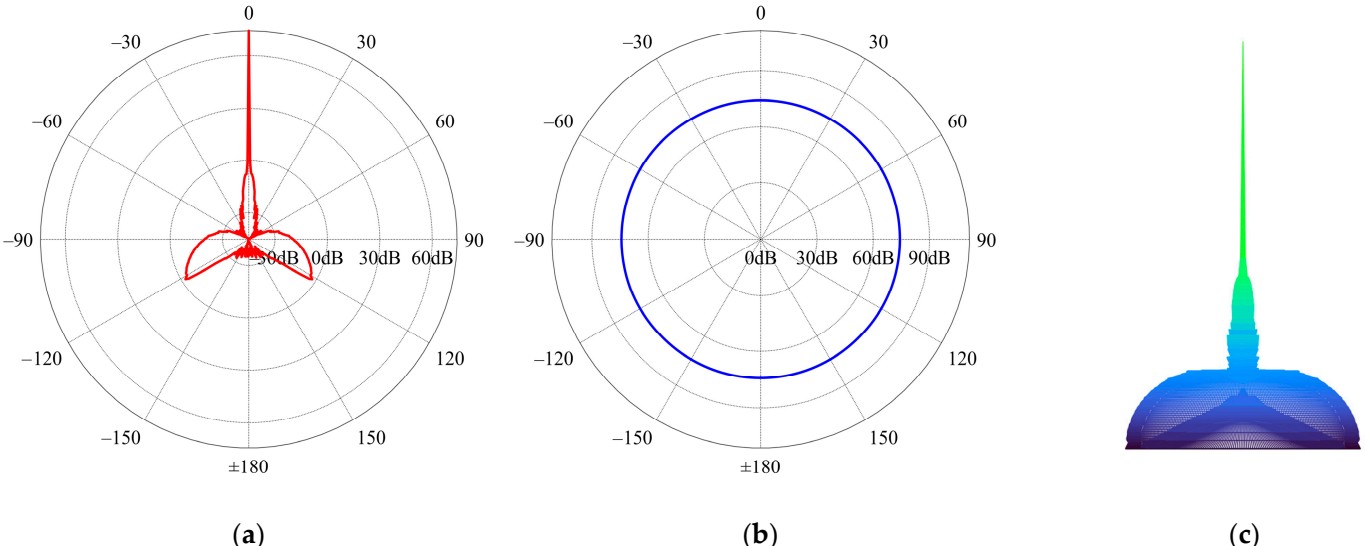

**(a)**                                                       **(b)**                                                       **(c)**

**Figure 3.** The typical radiation pattern of the FAST antenna at 1905 MHz: (**a**) 2D-vertical direction (zenith angle is 0°); (**b**) 2D-horizontal direction (elevation angle is 0°); (**c**) 3D.

*2.2. IMT-2000 Base Station*

The IMT-2000 system was formally proposed by ITU in 2000, which extends the 2G communication system from narrowband to broadband. The IMT-2000 system is mainly committed to mobile internet technology, which fully integrates wireless communication and other communication technologies, such as the Internet. It operates in the 2000 MHz frequency band, with an uplink frequency band of 1890–2030 MHz and a downlink frequency band of 2110–2250 MHz, abbreviated as IMT-2000. The IMT-2000 system mainly adopts Code Division Multiple Access (CDMA) as its core technology and has now formed three mainstream technology standards, including wideband W-CDMA, CDMA-2000, TD-SCDMA, and others. The emergence of the IMT-2000 system has made mobile communication services capable of multimedia transmission, better transmission quality, and higher spectral efficiency [16].

Wideband Code Division Multiple Access, referred to as W-CDMA, is a technology that evolved from the GSM core network. It primarily employs CDMA with a 5 MHz bandwidth, fast power control for both uplink and downlink, downlink transmit diversity, and synchronous and asynchronous operation between base stations. W-CDMA is the most widely adopted, has the most diverse range of terminal equipment among the IMT-2000 standards, and is currently operated mainly by CUCC. CDMA-2000 is a wideband CDMA technology developed from the narrowband CDMA-2000 technology. The forward link can adopt transmit diversity and forward/backward power control to improve channel fading resistance and channel capacity. It is fully compatible with the CDMA-2000 system and inherits the transmission characteristics of the CDMA-2000 system. It is currently operated by CTCC. TD-SCDMA technology was proposed by China to ITU in 1999. It

integrates intelligent wireless, synchronous CDMA, software-defined radio, and other related technologies. It has significant advantages in terms of business support, spectrum utilization, switching efficiency, anti-interference, and compatibility. It is highly valued by communication equipment manufacturers and is primarily operated by CMCC, as shown in Figure 4.

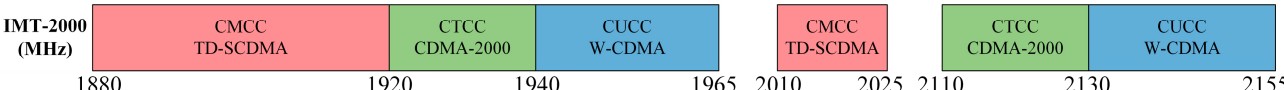

**Figure 4.** The frequency distribution of each operator (IMT-2000 frequency band).

The frequency bands used in the FAST RQZ for IMT-2000 include four frequency bands, namely, 1880–1920 MHz, 2010–2025 MHz, 2110–2130 MHz, and 2130–2155 MHz. In the analysis of interference characteristics of FAST-IMT, the EMI from the base station, which serves as the transmitting end, consists of two parts: electromagnetic leakage interference from the transmitter equipment and radiation interference from the IMT-2000 base station antenna [2]. Compared with the radiation interference from the base station antenna, the leakage interference from the transmitter equipment is weaker. Therefore, when analyzing the EMI of base stations, the radiation interference from the base station antenna is primarily considered. According to the interference prediction method described in Section 3.1, to predict the interference generated by communication base stations, relevant parameters such as the baseband resource reference power of the base station antenna, antenna gain, antenna height, and antenna radiation pattern need to be taken into account. All IMT-2000 base stations in the FAST RQZ use the same antenna model, which operates in 880–960/1700–2200/1880–2700 MHz frequency bands, with an azimuth of 65° and a tilt angle of 0°. Figure 5 shows the typical directional patterns of the IMT-2000 base station antenna in the working frequency bands, including the central horizontal plane (elevation angle is 0°), main vertical plane (zenith angle is 0°), and three-dimensional radiation pattern. The base station antenna exhibits a relatively wide main beam with a low gain but also shows significant sidelobes that may contribute to cross-polarization and intermodulation interference. Additionally, the minimum gain of the antenna is −30 dB, and the peak gain is 0 dB.

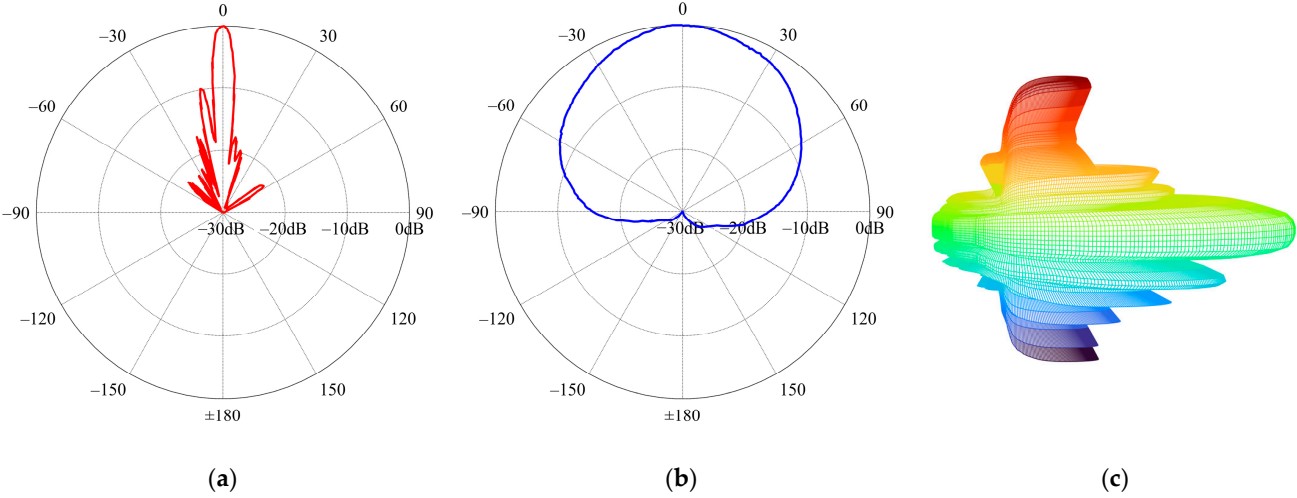

(**a**)          (**b**)          (**c**)

**Figure 5.** *Cont.*

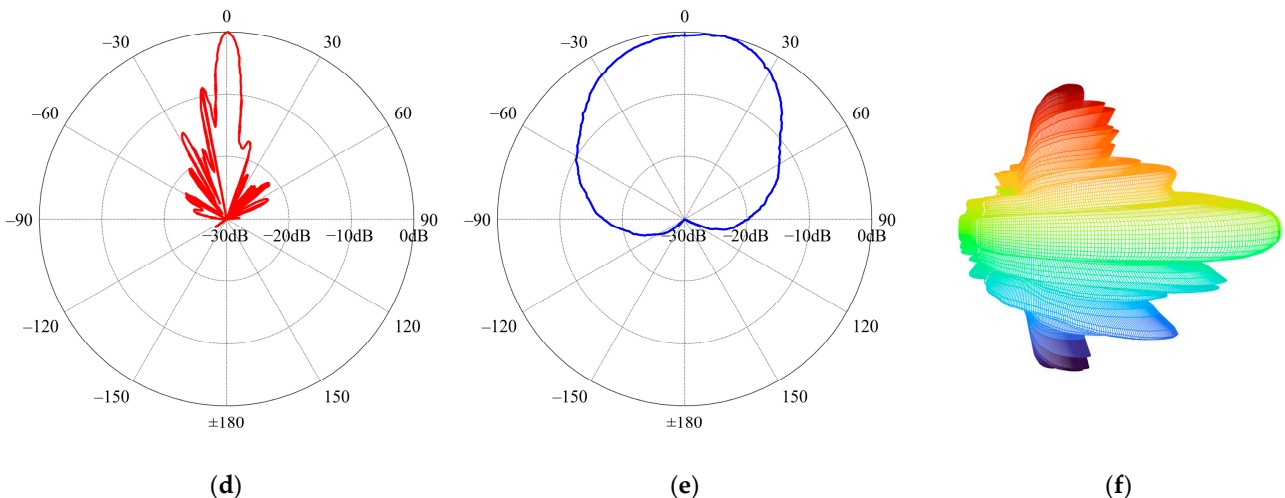

(**d**)　　　　　　　(**e**)　　　　　　　(**f**)

**Figure 5.** The typical radiation pattern of the base station antenna at 1700–2200 MHz and 1880–2700 MHz: (**a**) 2D-vertical direction (1700–2200 MHz); (**b**) 2D-horizontal direction (1700–2200 MHz); (**c**) 3D (1700–2200 MHz); (**d**) 2D-vertical direction (1880–2700 MHz); (**e**) 2D-horizontal direction (1880–2700 MHz); (**f**) 3D (1880–2700 MHz).

## 3. EMI Analysis

### 3.1. Analysis Method

EMI may exist at any time due to one or more propagation mechanisms, and which propagation mechanism is dominant mainly depends on the current weather conditions, the terrain environment, the radio frequency used, the percentage of time of interest, and the propagation path distance, among others. In the problem of interference prediction, it is difficult to propose a unified and consistent method for different distances and time percentages. Therefore, to ensure prediction accuracy, appropriate propagation models should be selected for different climate and path conditions [17].

In radio astronomy, the power *I* received from the interfering source during observation time *T* can be expressed as [18]

$$I = \frac{1}{N}\sum_{i=1}^{N}\frac{P_t(i)\cdot G_t(i)\cdot G_r(i)}{L_p(i)} \tag{1}$$

where $N$ is the number of samples in the $T$; $P_t(i)$ is the transmit power level value (W) of the radio astronomy input antenna at instant $i$; $G_t(i)$ is the gain of the interference source antenna in the direction of the receiving antenna at instant $i$; $G_r(i)$ is the gain of the receiving antenna in the direction of the transmitter at instant $i$; $L_p(i)$ is the transmission loss at instant $i$. Here, $N$, $P_t(i)$, $G_t(i)$, and $G_r(i)$ are all set parameters, and $L_p(i)$ can be predicted by the corresponding propagation model.

In the interference prediction analysis of FAST-IMT, ITU-R P.2001 is employed to predict the basic transmission loss caused by signal enhancement and fading. This model can be over distances ranging from 3 km to at least 1000 km beyond the effective range of 0% to 100% annual coverage [19]. The prediction analysis model involves four sub-models, namely, propagation close to the surface of the Earth, anomalous propagation, troposcatter propagation, and ionospheric propagation. Finally, a Monte Carlo simulation-based comprehensive method is utilized to predict the total transmission loss along the path, as shown in Figure 6.

(1)　Sub-model 1—Propagation close to the Earth's surface: when the height of the transmitting and receiving antenna is low to the ground, with the maximum radiation direction along the surface, radio waves mainly propagate along the surface of the Earth. At this time, the transmission loss mainly includes free-space basic transmission loss, diffraction loss, clear-air effect, atmospheric attenuation, and other factors.

(2)    Sub-model 2—Anomalous propagation: atmospheric propagation mainly refers to the anomalous propagation of the atmospheric layers, namely, the atmospheric waveguide phenomenon.

(3)    Sub-model 3—Troposcatter propagation: affected by different solar irradiation intensities, thus forming an uneven propagation medium in the troposphere and causing scattering. The troposcatter transmission loss mainly considers the basic troposcatter transmission loss, rain–snow precipitation attenuation, and atmospheric absorption loss.

(4)    Sub-model 4—Ionospheric propagation: for long-haul and low-frequency predictive links, it is essential to consider the ionospheric scattering transmission loss caused by sporadic-E, mainly including the one-hop propagation mode and two-hop propagation mode.

(5)    Set relevant parameters based on the device and environment: FAST and IMT-2000 base station locations, frequencies, heights of transceiver antennas, environment type, time probability, location probability, and other basic path information parameters. Then, determine the propagation parameters such as the propagation distance, path midpoint position, sea propagation length, path elevation angle, refractive index, precipitation probability, equivalent Earth radius, effective height, and path roughness parameters.

(6)    Free-space basic transmission loss: when the transmitting and receiving antennas are located within the "visible" distance from each other, the radio wave propagates point-to-point along a straight line without reflection or scattering.

(7)    When the transmitting and receiving antennas are beyond the line-of-sight range, radio waves propagate mainly through diffraction, including diffraction losses for the Earth's spherical surface, Bullington diffraction losses for the actual profile, and a notional smooth profile.

(8)    The primary components of the gaseous attenuation effect are total gaseous attenuation occurring during non-rain periods and gaseous attenuation resulting from water vapor in non-rain and rain situations.

(9)    The clear-air zero-fade effect under no atmospheric ducting mainly includes the refractive index change effect, the reflectivity of rain clouds, and the atmospheric thermal noise.

(10)    The transmit angle-dependent loss is the unique angular attenuation in the irregular propagation mechanism. When the corrected path angular separation is not greater than 0, its loss is 0; otherwise, it is the product of angular attenuation and the corrected total angular separation.

(11)    The time-distance-dependent loss depends on the distance and time percentage of the great circle.

(12)    According to the longitude and latitude coordinates of the station site, combined with the climatic zone model specified by ITU, determine the climatic zone and obtain the climatic zone's meteorological and atmospheric structure parameters.

(13)    Rain–snow and precipitation attenuation loss include the attenuation caused by rain, snow, and rainfall in the typical path between the transmitting and receiving antennas. It can be calculated using iterative functions based on the positions of the transmitting and receiving terminals, the antenna heights, and the path length between the transmitting and receiving antennas.

(14)    The dominant factor contributing to atmospheric absorption loss is the combined effect of total gaseous attenuation during non-rain periods and gaseous attenuation resulting from water vapor in both non-rain and rainy situations.

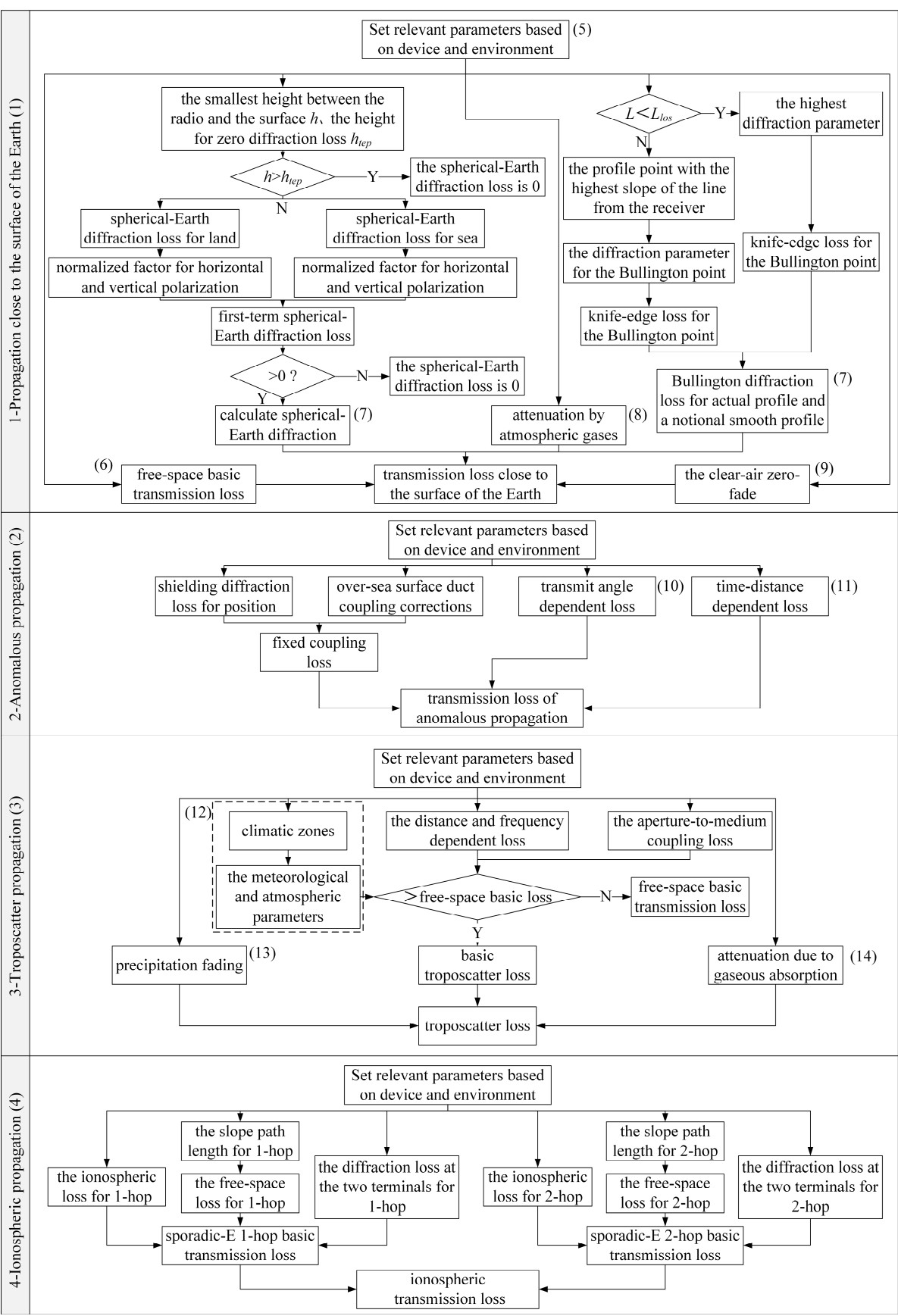

**Figure 6.** The propagation prediction process of the ITU-R P.2001 model.

*3.2. Analysis Result*

Using the interference evaluation method described in Section 3.1, and based on the 3 arc-seconds 90 m (SRTM3) high-precision elevation data, the interference caused by the IMT-2000 base stations listed in Figure 1 regarding FAST is analyzed. Because the zenith angle of FAST will change with different observation tasks and the antenna gain towards the direction of receiving signals from the base station will also change, in order to consider all possible scenarios to the maximum extent in the calculation process, several special parameters are set, as follows:

- Frequency: The upper and lower limits of each IMT-2000 base station working frequency band are selected as the analysis main frequency points, including multiple frequencies such as 1880 MHz, 1885 MHz, 1900.4 MHz, 1905 MHz, 1910.4 MHz, 1920 MHz, 2010 MHz, and 2025 MHz under CMCC and 2130.1 MHz, 2131.1 MHz, 2135.1 MHz, 2136.1 MHz, and 2155.1 MHz under CUCC.
- Time percentage: Five kinds of time probability, including 1%, 10%, 50%, 90%, and 99%.
- FAST receiving antenna zenith angle: 0°, 10°, 20°, 30°, and 40°.
- Antenna radiation direction: towards the base station transmitting antenna and away from the base station transmitting antenna.

Because the working frequency of the FAST receiver does not involve 1900–2000 MHz, the communication base station operating in 1900.4 MHz, 1905 MHz, 1910.4 MHz, and 1920 MHz under CMCC will not cause interference regarding FAST. Here, we only analyze the communication base station that shares the working frequency band with the receiver.

Based on the above parameters, we analyzed the interference situation of IMT-2000 mobile base stations, with a total of 62,415 data analyzed, of which 34,521 data meet the requirements, accounting for 55.31%. The evaluation results are classified and counted according to nine distance ranges of 5–6 km, 6–7 km, 7-8 km, 8-9 km, 9-10 km, 10-15 km, 15–20 km, 20–25 km, and over 25 km, as shown in Figure 7 (Note: there are no IMT-2000 base stations of CTCC near the FAST). The results show that: under the current conditions, the proportion of data meeting the FAST threshold requirements for base stations within a distance range of over 25 km is the highest, accounting for 64.03%, while the proportion for the 7–8 km range is the lowest, accounting for only 24.21%. Meanwhile, Table 1 provides the statistical proportion of data that meets the requirements for different receivers of FAST involved in the IMT-2000 frequency band. In the IMT-2000 frequency band, the operating frequency of the communication base station of CUCC is above 2100 MHz, which does not involve the B05 (1100–1900 MHz) receiver. Within the frequency band of the B05 receiver, the analysis data of IMT-2000 base stations meet the FAST protection requirements, accounting for 55.06%. The percentage is 56.00% in the B07 receiver frequency band.

**Table 1.** The statistical analysis of the proportion of IMT-2000 receivers meeting the FAST interference threshold.

| Receiver | CMCC | CUCC | Total Data | Proportion |
|---|---|---|---|---|
| B05 (1100–1900 MHz) | 25,298 | - | 45,945 | 55.06% |
| B07 (2000–3000 MHz) | 400 | 8823 | 16,470 | 56.00% |
| Total | 25,698 | 8823 | 62,415 | 55.31% |

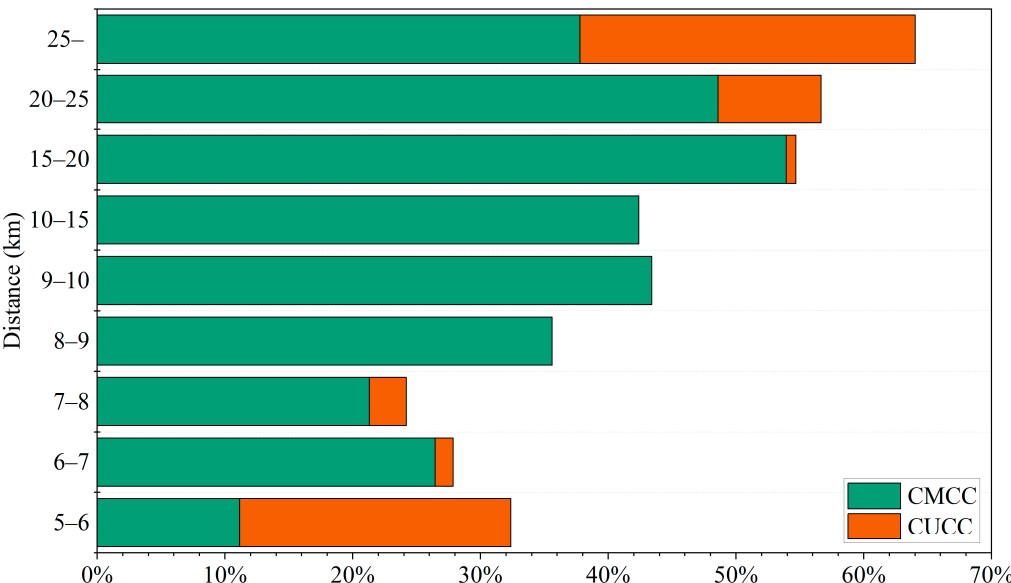

**Figure 7.** The statistical graph of the proportion of IMT-2000 base stations meeting the FAST interference threshold at various distances.

To suppress the EMI of communication base stations on FAST when the received power exceeds the interference threshold, protective measures can be implemented from the interference source. The most direct and effective measure is to shut down the communication base station, but this method results in the loss of communication quality and has a high cost [20]. In addition, Wang et al. provide four effective measures for mitigating the interference of communication base stations. These measures include increasing the operating frequency of the base station, adjusting the radiation direction of the base station antenna, reducing the height of the base station antenna installation, and decreasing the radiation power of the base station [12]. By implementing these measures, the interference received power of the base station at FAST can be significantly reduced while ensuring communication efficiency and quality.

*3.3. Result Validation*

In order to verify the applicability and accuracy of the deterministic radio wave propagation model used in this article, based on the frequency parameters of the IMT-2000 base station, the radio propagation characteristics of the propagation link formed between the experimental test base station and the surrounding receiving test positions are predicted. Testing was conducted using calibrated testing equipment and was in accordance with standards such as GB/T 12572-2008, HJ/T 10.2-1996, and GJB2080, and the received power at the receiving point is measured and compared with the simulation prediction results of the propagation model.

Four IMT-2000 base stations, including test base stations 1, 2, 3, and 4, were selected at different directions and distances around FAST. These base stations were operated by different operators and served as test sites. Then, multiple receiving points were set up in different sectors. In order to ensure the accuracy of the test results and avoid interference from terrain, buildings, high-voltage power lines, and other factors, the receiving antenna should be located in flat and open areas. The relationship between the test station and the various receiving test points, as well as the installation of the transmitting and receiving equipment, are shown in Figure 8; the notation "SEC120_1" in the figure refers to the first selected receiving point in sector 120 of the base station, and so on. The specific parameters of the field test are shown in Table 2.

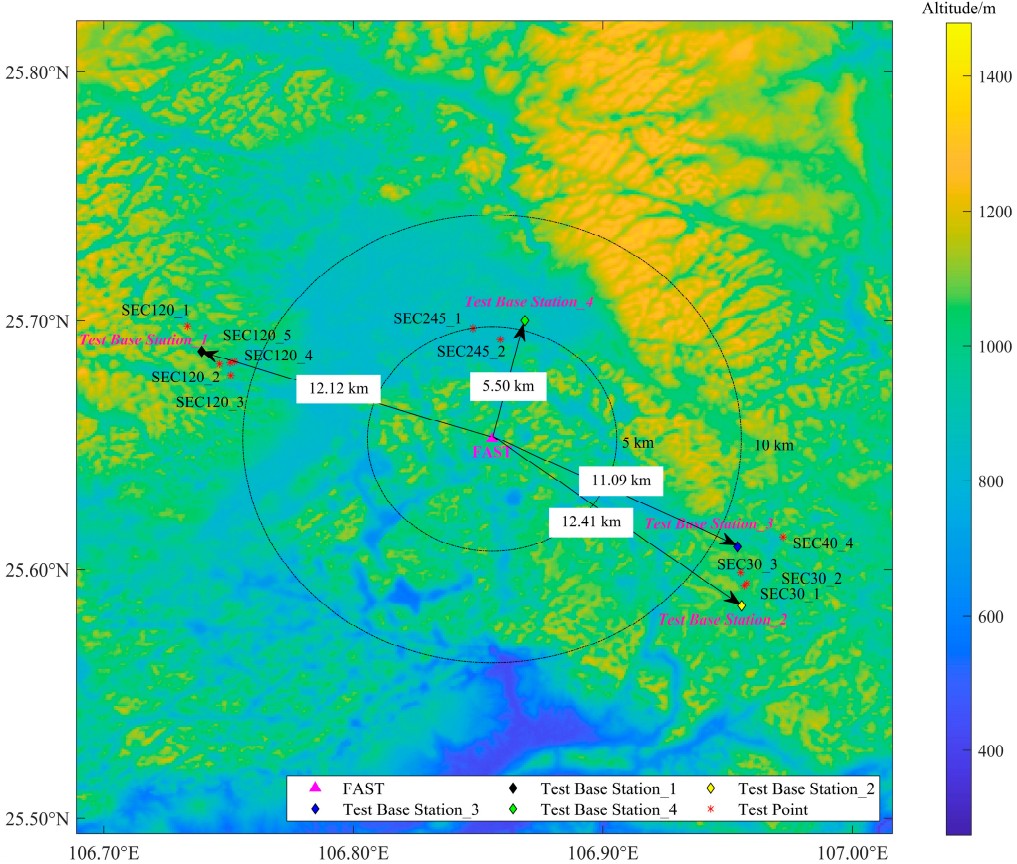

**Figure 8.** The location information for IMT-2000 test stations.

**Table 2.** The field test parameter for the IMT-2000 frequency band.

| Test Station Parameters | | | | | | | Receiver Point Parameters | |
|---|---|---|---|---|---|---|---|---|
| **Test Station** | **Location** | **Altitude (m)** | **Antenna Height (m)** | **Distance from FAST (km)** | **Frequency (MHz)** | **Transmit Power (W)** | **Antenna Height (m)** | **Path Distance (km)** |
| 1 | (106.74° E, 25.69° N) | 1150 | 15 | 12.12 | 1895/1909.4 | | | |
| 2 | (106.96° E, 25.59° N) | 1035 | 12 | 12.41 | 1895 | 20 | 4 | 0.9–2.1 |
| 3 | (106.95° E, 25.61° N) | 960 | 21 | 11.09 | 1895 | | | |
| 4 | (106.87° E, 25.70° N) | 930 | 15 | 5.50 | 1895 | | | |

Taking test base stations 1, 2, 3, and 4 as the main test base stations, we carried out the radio wave propagation characteristics prediction analysis and testing for the IMT-2000 frequency band mobile communication signals of different operators. The statistical results of the predicted link radio wave propagation characteristics between each test station and the surrounding test points are shown in Figure 9 and Table 3. After comparative analysis, the results show that: the predicted results are consistent with the measurement results, with a consistent overall trend. Table 3 shows the mean error (ME), mean deviation (MD), and standard deviation (SD) of four different tests conducted on the base station. Based on the results, test base station 1 had the largest ME of −4.23 dB, indicating a significant underestimation of the received signal power. Test base station 3 had the largest MD and SD values of 10.00 dB, which suggests that the errors for this base station were consistently large and highly variable. The last row presents the total measurement and prediction errors, with an ME of −1.20 dB, an MD of 4.43 dB, and an SD of 5.83 dB. Overall, the total

ME value for all tests is negative, while the MD and SD values are positive, indicating an overall underestimation of the base station performance with relatively good accuracy and variability. However, due to the proximity of several test points near buildings and high-voltage lines near the test base station 3, only the SEC40_4 receiving point was selected. As a result of the small number of test samples, a significant error was observed for the test base station 3.

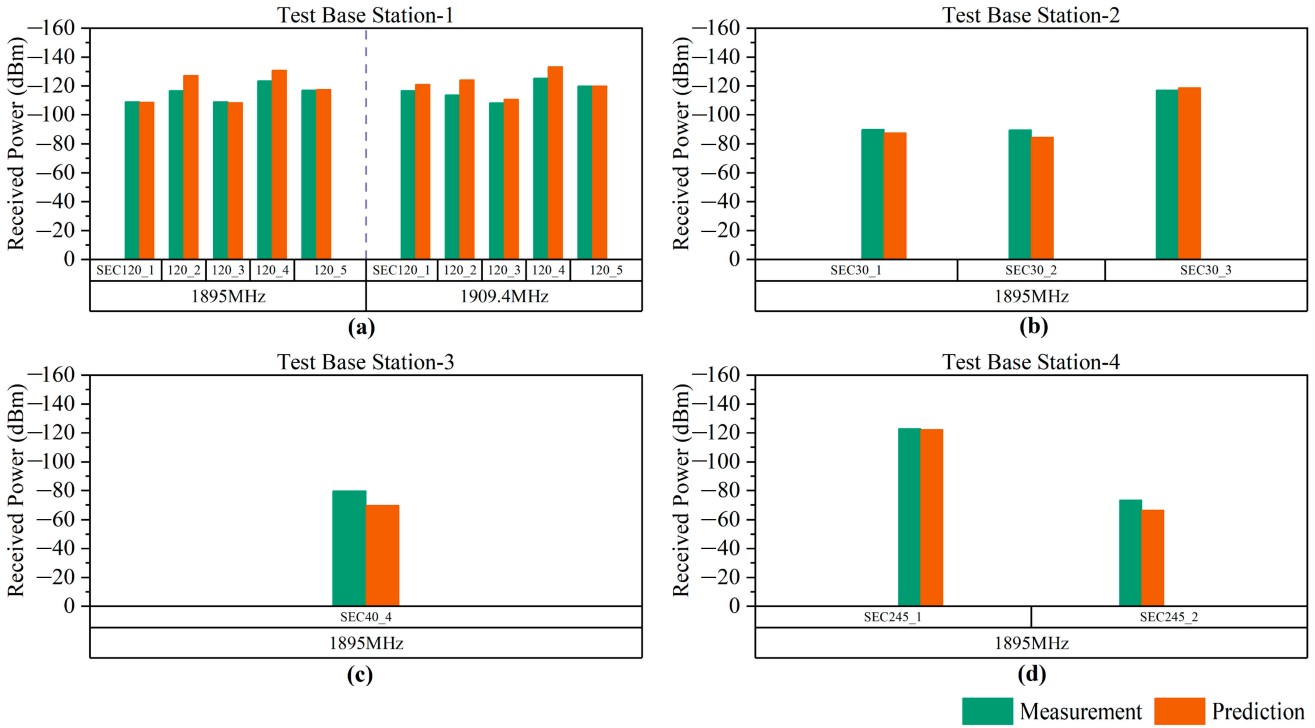

**Figure 9.** The comparative analysis and statistics between the measurement and prediction: (**a**) Test base station-1; (**b**) Test base station-2; (**c**) Test base station-3; (**d**) Test base station-4.

**Table 3.** The error analysis of the measurement and prediction.

| Test Base Station | ME (dB) | MD (dB) | SD (dB) |
|---|---|---|---|
| 1 | −4.23 | 4.49 | 6.05 |
| 2 | 1.93 | 2.87 | 3.23 |
| 3 | 10.00 | 10.00 | 10.00 |
| 4 | 3.65 | 3.65 | 4.82 |
| Total | −1.20 | 4.43 | 5.83 |

Therefore, the interference evaluation model used in this article can effectively support the analysis of the radio wave propagation characteristics of the link between the IMT-2000 base stations and FAST in the region.

## 4. Conclusions

This article analyzes whether the IMT-2000 base station in the FAST RQZ affects the normal observation operation of FAST, based on the frequency parameters of FAST and IMT-2000 mobile communication base stations using the ITU-R P.2001 method verified by experiments. By comparing the radio astronomy protection requirements and the protection threshold of FAST, it is concluded that the IMT-2000 base stations in the FAST RQZ meet the threshold data, accounting for only 55.31%. The proportion of data meeting the FAST threshold requirements for base stations within a distance range of over 25 km is the highest, accounting for 64.03%, while the lowest proportion is only 24.21% for the 7–8 km range. Furthermore, the analysis results validated that the predicted ME is −1.20 dB, the MD is

4.43 dB, and the SD is 5.83 dB. Furthermore, the ITU-R P.2001 selected in this article is an ITU recommendation. The universal method can be analyzed more accurately by combining radio meteorological parameters in different regions. The significance of adopting the universal method in this article extends beyond planning mobile communication services in the FAST RQZ and improving the electromagnetic environment around FAST. It also serves as a valuable supplementary approach for other frequency-dependent devices. In the future, the combination of artificial intelligence and other methodologies will be employed to develop localized statistical analysis methods, aiming to provide more robust support for evaluating relevant business operations.

**Author Contributions:** Conceptualization, J.W. and C.Y.; methodology, software, and validation, J.W., Y.Z., Y.S. and C.Y.; formal analysis, Y.H.; data curation, Y.S. and C.Y.; writing—original draft preparation, Y.Z.; writing—review and editing, J.W., C.Y., Y.S., H.Z., J.S. and D.L.; supervision and project administration, J.W. All authors have read and agreed to the published version of the manuscript.

**Funding:** This research received no external funding.

**Institutional Review Board Statement:** Not applicable.

**Informed Consent Statement:** Not applicable.

**Data Availability Statement:** Not applicable.

**Acknowledgments:** Part of this work was supported by the National Astronomical Observatories, the Chinese Academy of Sciences, and the Radio Regulatory Commission of Guizhou Province, China.

**Conflicts of Interest:** The authors declare no conflict of interest.

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
