# Peer review of "The Analysis and Verification of IMT-2000 Base Station Interference Characteristics in the FAST Radio Quiet Zone"

_universe, doi:10.3390/universe9060248_

Round 1
Reviewer 1 Report
This paper describes in detail the radio environment at the FAST telescope, but does not show any new statistical methods.
Author Response
Response to Reviewer 1 Comments
Point 1: This paper describes in detail the radio environment at the FAST telescope, but does not show any new statistical methods
Response 1: Thank you very much for your advice. We are sorry we did not make that clear in the original manuscript. The main focus of this article is to analyze the electromagnetic interference status of IMT-2000 frequency band communication base stations at FAST. The analytical model employed in this study utilizes the ITU-R P.2001 deterministic propagation model as recommended by ITU. Additionally, this model has been validated through our field testing in the terrain areas surrounding FAST, which supports the accuracy of our research findings. In the future, we will combine artificial intelligence and other methods to form localized statistical methods. Following your comment, we have made the following modifications:
Lines 18-27: In this study, we aim to analyze the electromagnetic interference (EMI) to Five-hundred-meter Aperture Spherical radio Telescope (FAST) caused by base stations in the International Mobile Telecommunications-2000 (IMT-2000) frequency band. By analyzing the frequency bands used by the transmitting and receiving devices and the surrounding environmental parameters and utilizing an approach to predicting radio wave propagation loss that is based on deterministic methods, we conclude by comparing the predicted received power at the FAST with its interference protection threshold. Our analysis demonstrates that currently, only 55.31% of IMT-2000 base stations in the FAST radio quiet zone (RQZ) meet the protection threshold. Additionally, this article verifies the applicability and accuracy of the radio wave propagation model used in the research based on field strength measurements.
Lines 374-376: In the future, the combination of artificial intelligence and other methodologies will be employed to develop localized statistical analysis methods, aiming to provide more robust support for evaluating relevant business operations.
Reviewer 2 Report
I have two main comments/suggestions.
1. The authors should discuss the generalizability of their methodology and results to other radio telescopes that are dealing with interference (e.g., SKA, Greenbank, Chime, EVLA) etc.
2. The others do not come to a conlcusion recommendation regarding the science impact of the interference they model/observe for FAST nor what mitigation should or could occure. This should be discussed.
Finally I note that Arecibo has been decommissioned following hurricane damage and is not operational nor is it planned at this time to be rebuilt.
The English is understandable as is though an English native editor could help to improve the English.
Author Response
Response to Reviewer 2 Comments
Point 1: The authors should discuss the generalizability of their methodology and results to other radio telescopes that are dealing with interference (e.g., SKA, Greenbank, Chime, EVLA) etc.
Response 1: Thank you very much for your comments. We are sorry we did not make that clear in the original manuscript. Regarding the general applicability of the methods proposed in this article, we mainly considered that: (1) The ITU-R P.2001 selected in this article is an ITU recommendation. The universal method can analyze radio meteorological parameters in different regions more accurately. (2) FAST in the karst area is selected, and the terrain is complex, the prediction results of the chosen method have been verified in multiple links, proving the effectiveness of the method. Following your comment, we have added sentences to the article.
Lines 29-30: It also provides corresponding analysis methods and useful suggestions for analyzing electromagnetic radiation interference in other radio telescopes.
Lines 368-370: Furthermore, The ITU-R P.2001 selected in this article is an ITU recommendation. The universal method can be analyzed more accurately by combining radio meteorological parameters in different regions.
Point 2: The others do not come to a conclusion recommendation regarding the science impact of the interference they model/observe for FAST nor what mitigation should or could occure. This should be discussed.
Response 2: Thank you very much for your comments. We are sorry we did not make that clear in the original manuscript. Following your comment, we have added the following modifications in Section 3.2 Analysis Results:
Lines 300-310: To suppress the EMI of communication base stations on FAST when the received power exceeds the interference threshold, protective measures can be implemented from the interference source. The most direct and effective measure is to shut down the communication base station, but this method results in the loss of communication quality and has a high cost [20]. In addition, Wang et al. (2022) provide four effective measures to mitigate the interference of communication base stations. These measures include increasing the operating frequency of the base station, adjusting the radiation direction of the base station antenna, reducing the height of the base station antenna installation, and decreasing the radiation power of the base station [12]. By implementing these measures, the interference received power of the base station at FAST can be significantly reduced while ensuring communication efficiency and quality.
And we have added this reference as:
- Huang, S.J.; Zhang, H.Y.; Hu, H. Study on electromagnetic compatibility between FAST and mobile base stations. Jounal of Deep Space Exploration. 2020, 7, 144-151, doi:https://doi.org/10.15982/j.issn.2095-7777.2020.20190618004.
Point 3: Finally I note that Arecibo has been decommissioned following hurricane damage and is not operational nor is it planned at this time to be rebuilt.
Response 3: Thank you very much for your comments. We are sorry we did not make that clear in the original manuscript. The FAST activation time is short, and the continuous observation time may be longer. And the FAST is located in the karst mountain area surrounded by mountains, and may continue to operate in the future. Furthermore, This article adopts a general method. To explain more clearly, we have added sentences as follows:
Lines 38-40: FAST, known as the "Chinese Eye of the Sky", has a comprehensive performance about 10 times higher than the Arecibo telescope in the United States, which was destroyed in December 2020.
Lines 370-374: The significance of adopting the universal method in this paper extends beyond planning mobile communication services in the FAST RQZ and improving the electromagnetic environment around FAST. It also serves as a valuable supplementary approach for other frequency-dependent devices.
Reviewer 3 Report
Comments are referred to line numbers of the manuscript:
Lines 90 to 96: In Figure 1 the radius of each area is not described. From previous text paragraphs, for instance lines 46 and 47, it seems the radii are 5 kM, 10 km and 30 km. Please clarify it and, if possible, add this information in the Figure 1 text description. The quality of the image of this Figure 1 is bad. Even making a zoom of this image it is difficult to see CMCC and CUCC base stations locations and to distinguish them. Would be possible to use a better quality image and to put it wider to see the details without the need of a zoom view?
Lines 102 to 104: The interference threshold levels depicted in Figure 2 are not clear to me. For instance, it seems that input power levels (at the receiver?) are between -200 and -300 dBW. Is it correct? Instead the Figure 2, it will be much clear if threshold values are written in a Table.
Lines 107 to 112: Text in lines 107 to 109 is about radiation pattern of IMT-2000 base stations, but in Figure 3 there are plots about radiation pattern of FAST antenna. Please, change the sentence in lines 107 to 109 and refer it to FAST antenna. Concerning Figure 3 (a): what is the maximum value (it seems to be greater than 60 dB)? also in Figure 3 (b) why is the value also greater than 60 dB, for elevation angle 0º the FAST antenna gain should be much lower than the FAST antenna gain in vertical direction? and what are the gain levels in Figure 3 (c)?. This plot doesn't give more information than Figure 3 (a) plot. It seems to be the same. Please, clarify all.
Line 284: The sentence about the test equipment is not clear. Maybe you want to say that the experimental test equipment follows a standard (?).
Lines 296 and 297: What is the meaning of SEC30, SEC120 and 245P in Figure 8? It seems they are test receiving points for experimental tests, but this information appears later (line 309). It would be better to explain what are the acronyms SEC30, SEC120, etc. before (in lines 292 to 295). Also, the image quality is quite bad. It is difficult to read the words included in the map. It would be better to improve the image quality and to put it greater, maybe using the full width of the page.
Author Response
Response to Reviewer 3 Comments
Point 1: Lines 90 to 96: In Figure 1 the radius of each area is not described. From previous text paragraphs, for instance lines 46 and 47, it seems the radii are 5 kM, 10 km and 30 km. Please clarify it and, if possible, add this information in the Figure 1 text description. The quality of the image of this Figure 1 is bad. Even making a zoom of this image it is difficult to see CMCC and CUCC base stations locations and to distinguish them. Would be possible to use a better quality image and to put it wider to see the details without the need of a zoom view?
Response 1: Thank you very much for your comment. We are sorry we overlooked this detail in the original manuscript. Following your comments, we have added radius labels of 5 km, 10 km, and 30 km for the quiet zone in Figure 1, as well as enhanced the image clarity for greater clarity.
Figure 1. The distribution of IMT-2000 base stations around FAST.
Point 2: Lines 102 to 104: The interference threshold levels depicted in Figure 2 are not clear to me. For instance, it seems that input power levels (at the receiver?) are between -200 and -300 dBW. Is it correct? Instead the Figure 2, it will be much clear if threshold values are written in a Table.
Response 2: Thank you very much for your comment! We are sorry we did not make that clear in the original manuscript. To provide a more clear and visual representation of the interference threshold for FAST, we have added data labels to Figure 2. Additionally, we also added the text description.
Lines 104-108: Furthermore, in the FAST-IMT interference analysis, we are particularly concerned with the maximum input power that the FAST can tolerate, and the B05 and B07 receiver protection power thresholds are both -199 dBW.
Figure 2. The interference protection requirements for FAST.
Point 3: Lines 107 to 112: Text in lines 107 to 109 is about radiation pattern of IMT-2000 base stations, but in Figure 3 there are plots about radiation pattern of FAST antenna. Please, change the sentence in lines 107 to 109 and refer it to FAST antenna. Concerning Figure 3 (a): what is the maximum value (it seems to be greater than 60 dB)? also in Figure 3 (b) why is the value also greater than 60 dB, for elevation angle 0º the FAST antenna gain should be much lower than the FAST antenna gain in vertical direction? and what are the gain levels in Figure 3 (c)?. This plot doesn't give more information than Figure 3 (a) plot. It seems to be the same. Please, clarify all.
Response 3: Thank you very much for your comments. We are very sorry that we made a mistake. In this revised manuscript, we have corrected it. Figure 3(a) is the vertical plane in which the zenith angle is 0°, and the maximum value is 74.21 dB (1905MHz). Figure 3(b) shows the horizontal pattern of the FAST antenna when the elevation angle is 0°, which is an omnidirectional radiation pattern, and the radiation power is 74.21 dB. Since the horizontal plane is an omnidirectional radiation pattern, the three-dimensional radiation pattern is obtained by rotating 360° around the central axis in Figure 3(a) and Figure 3(c).
And we revised and added these sentences in Lines 112-120 as follows:
The typical antenna radiation patterns of the FAST antenna, include the main vertical plane (zenith angle is 0°), the central horizontal plane (elevation angle is 0°), and the three-dimensional radiation pattern, as shown in Figure 3. The pattern of the antenna reveals a well-shaped main lobe with a narrow beamwidth, with a peak gain of 74.21 dB, accompanied by a number of minor lobes and back lobes of lower gain, with the highest side lobe located at about 120° with respect to the main lobe, as shown in Figure 3(a). Figure 3(b) shows that the horizontal pattern of FAST is directionless over 360° when the elevation angle is 0°, exhibiting a uniform distribution of equivalent radiated power with a gain of 74.21 dB.
Point 4: Line 284: The sentence about the test equipment is not clear. Maybe you want to say that the experimental test equipment follows a standard (?).
Response 4: Thank you very much for your comment. We are sorry we did not make that clear in the original manuscript. Your understanding is correct, and you want to say that the experimental test equipment follows some standards. To explain more clearly, we have added sentences as follows:
Lines 316-317: Testing was conducted using calibrated testing equipment and in accordance with standards such as GB/T 12572-2008, HJ/T 10.2-1996, GJB2080, …
Point 5: Lines 296 and 297: What is the meaning of SEC30, SEC120 and 245P in Figure 8? It seems they are test receiving points for experimental tests, but this information appears later (line 309). It would be better to explain what are the acronyms SEC30, SEC120, etc. before (in lines 292 to 295). Also, the image quality is quite bad. It is difficult to read the words included in the map. It would be better to improve the image quality and to put it greater, maybe using the full width of the page.
Response 5: Thank you very much for your suggestion. We are sorry we did not make that clear in the original manuscript. SEC is an abbreviation for "sector," where SCE30 and SCE120 denote 30 and 120 sectors, respectively. Additionally, there needed to be a correction in referring to SEC245 as 245P, which has been corrected in the paper. Following your comments, we have also revised Figures 8 and 9 to improve their clarity. And we have made a supplementary explanation in the article.
Lines 327-328: the notation "SEC120_1" in the figure refers to the first selected receiving point in sector 120 of the base station, and so on.
Figure 8. The location information for IMT-2000 test stations.
Figure 9. The comparative analysis and statistics between measurement and prediction.
